# MiR-125b-2 Knockout in Testis Is Associated with Targeting to the PAP Gene, Mitochondrial Copy Number, and Impaired Sperm Quality

**DOI:** 10.3390/ijms20010148

**Published:** 2019-01-03

**Authors:** Longlong Li, Yanling Zhu, Ting Chen, Jiajie Sun, Junyi Luo, Gang Shu, Songbo Wang, Xiaotong Zhu, Qingyan Jiang, Yongliang Zhang, Qianyun Xi

**Affiliations:** Guangdong Provincial Key Laboratory of Animal Nutrition Control, College of Animal Science, South China Agricultural University, Guangzhou 510642, China; longlongli0607@163.com (L.L.); kerryju27@163.com (Y.Z.); allinchen@scan.edu.cn (T.C.); jiajiesun@scau.edu.cn (J.S.); luojunyi@scau.edu.cn (J.L.); shugang@scau.edu.cn (G.S.); songbowang@gmail.com (S.W.); xtzhu@scau.edu.cn (X.Z.); qyjiang@scau.edu.cn (Q.J.)

**Keywords:** miR-125b-2, testis, PAP, reproduction, sperm, mitochondria

## Abstract

It has been reported that the miR-125 family plays an important role in regulating embryo development. However, the function of miR-125b-2 in spermatogenesis remains unknown. In this study, we used a model of miR-125b knockout (KO) mice to study the relationship between miR-125b-2 and spermatogenesis. Among the KO mice, the progeny test showed that the litter size decreased significantly (*p* = 0.0002) and the rate of non-parous females increased significantly from 10% to 38%. At the same time, the testosterone concentration increased significantly (*p* = 0.007), with a remarkable decrease for estradiol (*p* = 0.02). Moreover, the sperm count decreased obviously (*p* = 0.011) and the percentage of abnormal sperm increased significantly (*p* = 0.0002). The testicular transcriptome sequencing revealed that there were 173 up-regulated genes, including Papolb (PAP), and 151 down-regulated genes in KO mice compared with wild type (WT). The Kyoto Encyclopedia of Genes and Genomes (KEGG) and gene ontology (GO) analysis showed that many of these genes were involved in sperm mitochondrial metabolism and other cellular biological processes. Meanwhile, the sperm mitochondria DNA (mtDNA) copy number increased significantly in the KO mice, but there were no changes observed in the mtDNA integrity and mutations of mt-Cytb, as well as the mt-ATP6 between the WT mice and KO mice. In the top 10 up-regulated genes, PAP, as a testis specific expressing gene, affect the process of spermatogenesis. Western blotting and the Luciferase assay validated that PAP was the target of miR-125b-5p. Intriguingly, we also found that both miR-125b and PAP were only highly expressed in the germ cells (GC) instead of in the Leydig cells (LC) and Sertoli cells (SC). Additionally, miR-125b-5p down regulated the secretion of testosterone in the TM3 cell by targeting PAP (*p* = 0.021). Our study firstly demonstrated that miR-125b-2 regulated testosterone secretion by directly targeting PAP, and increased the sperm mtDNA copy number to affect semen quality. The study indicated that miR-125b-2 had a positive influence on the reproductive performance of animals by regulating the expression of the PAP gene, and could be a potential drugs and diagnostic target for male infertility.

## 1. Introduction

Infertility is one of the most common human health problems [1], and 40–50% of all cases are due to a male deficiency. In recent years, several clinical studies [2,3] have shown that male infertility is caused by sperm quality and impaired spermatogenesis. The process of spermatogenesis is highly sensitive to the fluctuations of the environment, and involves numerous endocrine and paracrine signals to coordinate the self-renewal and differentiation of spermatogonial stem cells [4]. Appropriate levels of reproductive hormones such as testosterone are a key factor of spermatogenesis [5]. Hormone screening can help define whether male infertility is caused by a gonadotropin deficiency. Spermatogenesis is regulated by thousands of genes. The molecular and cellular integrity of sperm cells are important for fertilization, while inappropriate gene expressions always cause disorders in spermatogenesis and fertility [6,7].

It has been widely reported that small RNA molecules, including small interfering RNAs (siRNAs) and microRNAs (miRNAs), have emerged as important regulators of gene expressions at the post-transcription or translation level [3]. Several miRNAs expressed abundantly in male germ cells, either throughout or during specific stages of spermatogenesis [8]. MiRNAs are a class of highly conserved small noncoding RNAs that primarily bind to complementary sequences in the 3′- untranslated region (UTR) of their target mRNAs, which results in mRNA degradation or repression of mRNA translation [9]. A study on the gonadotrope reports that the specific deletion of the dicer, an endoribonuclease involved in the biogenesis of miRNAs, leads to male and female infertility because the synthesis of the two gonadotropins is completely abolished [10,11]. Mature miR-125b originates from two precursors, miR-125b-1 and miR-125b-2. It is reported that miR-125b suppresses epithelial ovarian cancer cell migration and invasion [12], regulates p53 in embryonic stem cells [13] and plays an important role in lipogenesis [14]. Furthermore, studies have also shown that the miR-125 family plays a crucial role in regulating the zygotic genome activation in oocytes and embryos [15]. However, the molecular regulation of miR-125b-2 on male fertility remains unknown.

Previous studies in our lab found that all miR-125b-2 gene knockout (KO) mice appeared dysgenesis. This suggests that miR-125b-2 might be associated with animal reproduction. In our study, we used a miR-125b-2 knockout mice model to study the mechanism of miR-125-2 in reproductive performance, in order to provide a potential drugs and diagnostic target for male infertility.

## 2. Result

### 2.1. Phenotype of KO Mice

The KO mice were verified using tail DNA by PCR with primers 5′-ACATTACTGTAAGTTCTGATCTATA-3′ and 5′-GTACCGATTCTGAAGATTGTAT-3′. The sequences of wild type (WT) and KO mice were blasted (Figure 1). The seed sequence of miR125b-2 is CCCTGAGACCCTAACTTGTGAGGTATTTTAGTAACATCACAAG. Through statistical analysis, we found both the WT and KO male mice grew normally and had no difference in body weight and testis size (Table 1). The in vivo data suggested that there was no difference of body weight or testis weight when miR-125b-2 was knocked out in mice.

### 2.2. miR-125b-2 Knockout Causes Infertility

The effect and mechanism of miR-125b-2 on mouse reproductive performance was investigated. Male fertility was tested through four mating combinations, which included WT♀ × KO♂, KO♀ × KO♂, WT ♂ × KO♀, and WT♀ × WT♂. Comparing WT♂ × KO♀ with WT♀ × WT♂, the litter size of every gravida was significantly reduced (*p* = 0.011) and the percentage of non-parous females rose from 10% to 18%. Comparing WT♀ × KO♂ with WT♀ × WT♂, the litter size of every gravida was not significantly different (*p* = 0.978) and the percentage of non-parous females rose from 10% to 27%. But when we compared KO♀ × KO♂ with WT♀ × WT♂, the litter size of every nest decreased significantly (*p* = 0.0002), while the percentage of non-parous females rose from 10% to 38% (Figure 2A,B). For further exploration, we examined the mice’s sperm and found that the sperm count decreased in the KO mice (*p* = 0.011) compared with the WT group, as shown in Figure 2C. There was also a markable increase (*p* = 0.0002) in the percentage of abnormal sperms in the miR125b-2 knockout mice (Figure 2D). This result suggested that miR125b-2 could reduce the sperm quality and affect mouse fertility. To determine whether the reproductive hormone concentrations in the serum have changed between the WT and KO mice, we examined the levels of testosterone (T) and estradiol (E_2_) in the WT and KO serum. The WT group displayed an increase in testosterone concentration (*p* = 0.007) and a decrease in estradiol concentration (*p* = 0.02) compared with the KO group (Figure 2E,F). But a comparative histological analysis of the testis sections revealed that both the size and architecture of the tubular showed no significant difference in the KO mice, compared with the WT mice (Figure 2G). It is reported that testosterone is able to be converted to estradiol under the enzymatic actions of P450. When the enzyme activity is suppressed, testosterone cannot effectively be transformed into estradiol, which leads to focal hyperandrogenism [16]. Thus, to explain the mechanism of hormone changes, we quantified the testicular P450 expression. As expected, the KO displayed a lower P450 mRNA expression than the WT (*p* = 0.030) (Figure 2H). At the same time, we quantified the androgen receptor (AR) expression and found that no difference was exhibited in the AR mRNA expression compared with the WT. But the relative level of AR protein increased significantly in the KO mice (*p* = 0.012) (Figure 2I). It suggested that miR-125b-2 would affect the levels of sex hormones.

### 2.3. RNA-Seq Analysis Revealed the Changes in the Mouse Transcriptome between the KO and WT

The RNA samples from the testis of WT and KO mice were subjected to RNA-seq using an Illumina HiSeqTM 4000 instrument. A total of two samples were analyzed, which included the mixing of three samples for each condition. A total of 48,969,660 to 56,391,004 raw reads were generated in each library, and the valid data were 48,561,786 to 55,911,230 reads, with the valid ratios being more than 99% for the two libraries (the detail characteristics of transcriptome sequencing were showed in Appendix A). The quality evaluation indexes of the sequencing reads were verified by FastQC and RSeQC, which suggested that the results were reliable and met the requirements of the subsequent differentially expressed genes (DEGs) screen and data analysis.

At first, a fold change (FC) was detected to screen out the DEGs. The selection criteria were log2(FC) ≥ 1 or log2(FC) ≤ −1. FC = (FPKM.KO_G) expression level/(FPKM.WT_G) expression level. log2(FC) ≥ 1 refers to the up-regulated gene after miR-125b-2 knockout, while log2(FC) ≤ −1 refers to the down-regulated gene. Secondly, the significant differences’ genes (*p* < 0.05) were screened using Ballgown [17]. Finally, the *p*-value was corrected using the false discovery rate (FDR) [18] for multiple comparisons.

Among the two libraries, 32,667 genes were detected. Between the WT_G and KO_G libraries, 324 significant DEGs (*p* < 0.05) composed of 173 up-regulated and 151 down-regulated genes were identified (Appendix A). Firstly, we selected DEGs with FC ≥ 2 and *p* < 0.05. Then, according to the order of the *p*-value from small to big, the 10 genes with up-regulation and down-regulation were selected, respectively (Appendix A). To determine the function of the DEGs, we mapped them both in the GO database (Appendix A) and KEGG Pathway database (Appendix A). Among them, the most important biological processes included the mitochondrial metabolism, sperm chromatin condensation and other cellular biological processes. To confirm the accuracy of the RNA-seq results, the expressions of nine DEGs were analyzed using qRT-PCR (Appendix A). The qRT-PCR results suggested that the results of RNA-seq were reliable.

### 2.4. Sperm mtDNA Copy and Integrity

Interestingly, we also found that the DEGs of the WT and KO participated in the mitochondrial metabolism biological processes. The increase in the mitochondria DNA (mtDNA) copy number is also reported to decrease sperm quality. To determine the effect of miR-125b-2 on the mtDNA copy number in mouse sperm, we compared WT with KO mice. We detected the relative expression level of four marker genes’ mRNA, which stood for the mtDNA copy number, including ATPase6, COX2, Mit-1000, and mt-Cytb. The relative expression of mRNA was normalized by the detection of nuclear DNA. Compared with the WT group, the relative expression of ATPase6, COX2, Mit-1000 and mt-Cytb mRNA increased significantly in the KO group, with *p* = 0.01, *p* = 0.009, *p* = 0.004, *p* = 0.031, respectively (Figure 3A). The mtDNA copy number depends on the expression level of mt-TFA [19]. So, we next detected the mRNA level of mt-TFA. As expected, the mRNA level of mt-TFA increased significantly (*p* = 0.009) (Figure 3B). For the changes in the expression of the mitochondrial NADH dehydrogenase, we examined the mRNA level of ND1 and ND4 in the sperm. The results showed that the ND1 and ND4 increased significantly in the sperm of the KO mice (*p* = 0.007, *p* = 0.002). Consistent with the four marker genes’ results, as expected, the mRNA of the mt-TFA, ND1, and ND4 level increased significantly in the KO group (Figure 3C). To investigate both the mtDNA integrity and the mutations of mt-Cytb and mt-ATP6, we used primers to amplify the mtDNA-1, mtDNA-2, mt-Cytb, and mt-ATP6 sequences [20]. However, there was no change in the mutations of the mt-Cytb, mt-ATP6, and mtDNA integrity between the KO group and the WT group (Figure 3D,E).

### 2.5. miR-125b-5p Targeted Papolb (PAP) Gene to Suppress Testosterone Secretion in TM3 Cells

Among the top ten up-regulated genes according to the transcriptome results, PAP, a testis specific expressing gene, plays an important role in spermatogenesis. As the transcriptome results mentioned above, in the KO mice, the expression of PAP increased compared with the WT. Thus, we speculated whether the up regulation of PAP in the KO mice was related to miR-125b-2.

#### 2.5.1. miR-125b-5p Targeted on PAP

A flanking sequence analysis showed that the 3′-UTR of PAP mRNA contains a binding site that perfectly matched the seed region of miR-125b-5p. To verify that PAP was the target of miR-125b-5p, Chinese hamster ovary (CHO) cells were co-transfected with miR-125b-5p mimics/NC and pmirGLO-PAP, pmirGLO-PAP-Mut, and pmirGLO-PAP-Del. Forty-eight hours after transfection, the luciferase activity was assayed. The pmirGLO group showed the lowest luciferase activity when compared with the other groups (*p* = 0.00002). The reduction was rescued in the mutation group and deletion group (Figure 4A). The result of the luciferase activity showed that PAP was preliminarily the target gene of miR-125b-5p. To further validate whether miR-125b-5p was targeted on the PAP, we examined the expression level of PAP in the testicular tissue both of the KO and the WT. As expected, the qRT-PCR and Western blotting analysis showed that miR-125b-5p KO exhibited higher PAP mRNA expression (*p* = 0.029) and PAP protein expression (*p* = 0.023) than the WT (Figure 4B,C). These results collectively demonstrated that PAP was the target of miR-125b-5p.

#### 2.5.2. Both miR-125b-2 and PAP Affect the Secretion of Testosterone in TM3

To further confirm that miR-125b-2 could decrease the testosterone concentration (Figure 1E), we transfected miR-125b-5p mimics/inhibitors into the TM3 Leydig cell line. As expected, the miR-125b-5p mimics decreased the secretion of testosterone significantly (*p* = 0.024), while the miR-125b-5p inhibitors increased the secretion of testosterone markedly (*p* = 0.036) (Figure 5A). Also, we utilized PAP siRNA to explore the relationship between PAP and testosterone secretion. It is shown that PAP1 siRNA decreased the testosterone concentration of the TM3 supernatant (Figure 5B), which suggested that PAP played a significant in the testosterone secretion (*p* = 0.021).

#### 2.5.3. The Localization of miR-125b and PAP in Testicular Cells

To further validate the relationship between miR-125b and PAP, we isolated three types of testicular cells from the WT and KO mice. After purification of the isolated cells, we examined the expressions of PAP and miR-125b. The RT-PCR results showed that the miR-125b remarkably decreased in the germ cells (GC) (*p* = 0.0002), but showed no significant differences in Leydig cells (LC) (*p* = 0.093) and Sertoli cells (SC) (*p* = 0.49) (Figure 6A–C). Interestingly, we found that the PAP mainly expressed in GC, rarely expressed in LC, and could not be detected in SC (Figure 6F). Meanwhile, the results illustrated that the mRNA level of PAP remarkably increased in GC (*p* = 0.001) (Figure 6D), but there was no significant difference in LC (*p* = 0.108) (Figure 6E).

## 3. Discussion

Spermatogenesis is an intricate process of germ cell development and many genes participate in this process. Any defect in the gene expressions or regulations will disrupt spermatogenesis and cause male infertility [21]. Few studies have been conducted on the specific microRNA functions of spermatogenesis and male fertility [22,23]. During the breeding of the KO mice, we found that the litter size was significantly reduced and the infertility rate was significantly increased. Similar results have been reported that the miR-125 family plays an important role in regulating the maternal genes in the oocytes and embryos [15]. Through the statistical analysis of litter size and litter rate in the four mating combinations, we speculated that the knockout of miR-125b-2 could affect the female and male mouse reproduction performances. In this study, it was validated that miR-125b-2 had a great influence on the litter size of the female mice, which increased the infertility of the male mice. We chose to study the effect of miR-125b-2 on male mice because studies on the female mice are being done by other colleagues.

The testis is one of the most vital organs for animal reproduction and its development is strictly regulated by many processes. In this study, RNA-seq was used to identify the transcriptome differences of the testis between the the WT and KO mice. The transcriptome analysis identified 324 DEGs between the WT and KO groups, 173 up-regulated and 151 down-regulated. Among them, important biological processes, such as sperm chromatin condensation and mitochondrial metabolism, were included. This suggested that miR-125b-2 affected the spermatogenesis and sperm mitochondria.

Sperm quality is highly related to male infertility [24]. Our study revealed that compared with the WT group, the sperm count significantly reduced while the percentage of abnormal sperms significantly increased in the KO group, with a significant increase in the sperm mitochondria DNA copy number parallelly. Mitochondrion is a key organelle in sperm function under physiological conditions [25]. With one unit increase in the mtDNA copy number of mt-Cytb, there is a 4% decrease in the sperm swimming ability [26]. Others have also reported that an increase in the mtDNA copy number results in a decrease in the sperm motility. It has already been proven that changes in the transcription factor A, mitochondrial (TFAM) protein levels directly influence the mtDNA copy number. Heterozygous knockout mice showed a 50% reduction in the mtDNA copy number, whereas the TFAM protein showed a 50% increase in the mtDNA level in mice [27]. Interestingly, we found the TFAM mRNA level increased significantly with the increase of the mtDNA copy number in the KO mice, which was consistent with the study mentioned above. NADH dehydrogenase I (ND1) is used to determine mtDNA copy number, which has been reported to be a stable gene in a region of the genome that is rarely deleted [28]. NADH dehydrogenase 4 (ND4) is also commonly used, but is more susceptible to deletions because of their location [29]. Also, in our study, we found both the ND1 mRNA and ND4 mRNA of sperm increased significantly, which was consistent with the increase of the mtDNA copy number in the KO mice. These results collectively illustrated that the miR-125b knockout could reduce the sperm quality of male mice by impacting the mitochondrial function.

As the result of mRNA-seq, the PAP mRNA level increased significantly. It has been reported that the PAP, a cytoplasmic poly (A) polymerase, is responsible for the additional poly (A) tail extension of specific mRNAs in round spermatids [30,31,32], and encodes the transcriptional activator cAMP responsive element modulator (CREM), which is highly expressed in male germ cells [33]. To further confirm the function of PAP in the testis, we cultured GC, SC, and LC. Using RT-qPCR analysis, we found that PAP was mainly expressed in the germ cells, which was consistent with others’ reports. Others reported that there was a direct link between the deficiency of PAP and the arrest of mouse spermiogenesis [30]. Normal fertility and spermatogenesis allow for a 2- to 2.5-fold PAP overexpression in mice [31], and the overexpression of PAP interferes with cell growth and development [34,35]. Intriguingly, our results also support these demonstrations, that the overexpression of PAP can reduce the reproductive performance of male mice. PAP had a 4.2-fold overexpression in the KO compared with the WT, while the fertility of the KO male mice was decreased. The overexpression of PAP in Drosophila results in a dramatic elongation of mRNA poly (A) tails nonspecifically in the cell cytoplasm, which leads to the embryonic of lethality [35]. Our results are consistent with others’, because we also found that the overexpression of PAP could lower the reproductive performance. The regulatory mechanism between PAP and reproduction is worth studying because of its significant regulation function in reproduction.

Testosterone is produced in the LC in response to the luteinizing hormone (LH), and the appropriate secretion of testosterone, follicle-stimulating hormone, and LH are proved to be fundamentally important for normal spermatogenesis [36,37]. It is reported that the lowering of testosterone concentrations results in a decrease in overall sexual activity, thoughts, hot flushes, and fantasies. Male patients with androgen deficiency syndromes could take testosterone therapy and it effectively worked [38]. It is suggested that androgen resistance and spermatogenesis failure [39,40,41] can define whether male infertility is caused by a gonadotropin deficiency, and this will produce a bad effect on reproduction. In this study, the KO group showed that the testosterone concentration significantly increased while the estradiol concentration significantly decreased. These results implicated that the miR-125b-2 knockout mice might be under the androgen tolerant environment, which resulted in an increase of the infertility in male mice.

All of the stages of spermatogenesis are dependent on an intimate interaction between Sertoli cells (SC) and Leydig cells (LC), which provide a microenvironment essential for spermatogenesis [5]. Hormone screening helps define whether the reason for male infertility is a gonadotropin deficiency. Thus, we transfected miR-125b-5p mimics/inhibitor/PAP siRNA into TM3 cells to detect the difference in testosterone secretion. Interestingly, when the TM3 cells were transfected with miR-125b-5p mimics/PAP siRNA, the testosterone concentration of the TM3 cells’ supernatant decreased, while when transfected with miR-125b-5p inhibitors, the testosterone concentration of the TM3 cells’ supernatant increased. Western blotting and Luciferase assay further validated that miR-125b-5p decreased testosterone secretion in male mice testis by targeting PAP. 

In conclusion, our study is the first to demonstrate that miR-125b-2 decreases reproductive performance in male mice via both regulating testosterone secretion by targeting PAP and increasing the sperm mtDNA copy number. Our findings provide a better understanding of the molecular mechanism of infertility, and suggest that miR-125b-2 might be applied to clinical andrology and breeding management because of its significant role in spermatogenesis and fertilization.

## 4. Materials and Methods

### 4.1. Ethics Statement

The animal experiments complied with the guidelines of Guangdong Province on the Review of Welfare and Ethics of Laboratory Animals, approved by the Guangdong Province Administration Office of Laboratory Animals. All of the procedures were conducted according to the protocol (SCAU-AEC-2010-04-16) approved by the Animal Ethics Committee of South China Agricultural University.

### 4.2. Fertility Test

A MiR-125b-2 gene knockout model of mice was constructed by CRISPR/Cas9 System (Cyagen Biosciences Inc., USA). Fertility was assessed by mating the experimental mice. Those mice were allowed to mate for a 30-day period, and pairs were monitored regularly for signs of pregnancy. The non-pregnancy ratio was calculated by the number of non-pregnant female mice undergoing parturition over the total female mice in each group. The statistical analyses of the average litter size were performed with PASW Statistics 18 software.

### 4.3. Body Weight Measurement

Two groups (WT and KO) of 18–18.7 g male FVB mice were studied (eight animals/group). The mice were housed individually for one week to get acclimatized to the new environment. The body weight was measured every week following acclimatization for two months.

### 4.4. Morphological Observation of Testis

Four mice of every group were selected randomly for testicular histopathology. The left testis was fixed in 4% paraformaldehyde overnight, and the testis were washed with a 70% ethanol and Li2CO3-saturated solution. Following the standard procedures for paraffin-embedded tissues, the testis tissues were sliced into 5 μm sections and stained using hematoxylin-eosin staining (HE), according to a standard protocol for histopathologic analysis [42].

### 4.5. Sperm Quality Evaluation

The cauda epididymis was isolated after the mice were sacrificed, and was transferred into a small beaker containing 1 ml pre-warmed 37 °C normal saline, and was excised with fine scissors. After 10 min of incubation, the spermatozoa were allowed to swim out freely, and the undissolved tissues were removed. Aliquots of the sperm suspensions were prepared for the determination of sperm parameters, including count and morphology, which were evaluated as previously described [43]. Briefly, a 10-mL sperm suspension was analyzed to determine the spermatozoa count, using a hemocytometer. A 10-mL aliquot of sperm suspension was used to assess the sperm morphology. The suspension was uniformly smeared onto a glass slide, which was air-dried and fixed with methanol for 5 min. The spermatozoa were then stained with eosin for 1 h and washed with distilled water. The slides were air-dried at room temperature and the sperm morphology was examined using a light microscope to study 2000 sperm cells under a 400× magnification.

### 4.6. Target Prediction and Pathway Analysis

We predicted the target genes of miRNA in the mice at the genome level. In brief, the 3′-UTR sequences of the murine transcripts in the whole genome were obtained from Ensembl genome browser 80 (sscorfa10.2; http://www.ensembl.org/Sus_scrofa/). The mature differentially expressed miRNAs sequences were downloaded from miRbase release 21 (http://www.mirbase.org), and RNA hybrid software (http://www.bibiserv.techfak.uni-bielefeld.de/rnahybrid) was used to analyze the miRNA targets by using its own algorithm. Our prediction was restricted to a perfect match of the seed region (two–seven bases of the miRNA 5′ end; G:U matches were permitted), because of the importance of the seed sequence for miRNA–mRNA binding [44]. In addition, the matches were restricted to those with less than 220 kcal/mol of low free energy in the binding of miRNA–mRNA. Furthermore, the Database for Annotation, Visualization, and Integrated Discovery (DAVID) v6.7 online service (http://david.abcc.ncifcrf.gov/) [45] was used for gene ontology (GO) analysis, as well as the Kyoto Encyclopedia of Genes and Genomes (KEGG) pathway analysis based on all of the differentially expressed miRNA potential targets. Individual GO analysis and KEGG analysis of miR-125b-5p were also conducted based on the potential targets of miR-125b-5p.

### 4.7. Protein Extraction and Western blot

The testicular tissue was lysed with a RIPA buffer with protease inhibitors. The total soluble protein was quantified by the BCA protein assay. The total protein (30 μg) was loaded onto a 10% SDS-PAGE gel, separated by electrophoresis, and transferred onto a polyvinylidene difluoride membrane. The blots were blocked with 5% skim milk and incubated with a primary antibody overnight at 4 °C, followed by incubation with a secondary antibody for 1 h at room temperature, and measured with an Infrared Imaging System (LI-COR, Lincoln, NE, USA). The protein expression was normalized by the detection of β-actin (Abcam, Cambridge, UK). The protein from the testicular tissue was isolated to measure the expression level of PAP, and was probed using the primary monoclonal antibody (Santa Cruz, CA, USA).

### 4.8. Plasmid Construction, Transfection, and Luciferase Assay

To generate the mouse PAP 3′-UTR sequence containing the miR-125b target site CTCAGGG, two single-stranded DNA were synthesized (Sangon, Shanghai, China) and annealed to form a double-stranded DNA, which contained the recognition sites of the restriction enzymes XbaI and XhoI. The synthetic sequence was inserted into XbaI and XhoI enzyme-digested vector pGL3-Control (Promega Co., Madison, WI, USA), downstream of the luciferase gene. Meanwhile, the deleted and mutagenic PAP 3′-UTR reporter vectors were constructed with five exchanged nucleotides or a deleted target site in the same way. The sequences of three types of 3′-UTR were as follows: PAP 3′-UTR (sense, TCGAGTCTACTCTTTTACCAAGTCTCAGGGATACTATAAATTAGAGCTT; antisense, CTAGAAGCTCTAATTTATAGTATCCCTGAGACTTGGTAAAAGAGTAGAC), PAP 3′-UTR mut (sense, TCGAGTCTACTCTTTTACCAAGTGAGACGTATACTATAAATTAGAGCTT; antisense, CTAGAAGCTCTAATTTATAGTATACGTCTCACTTGGTAAAAGAGTAGAC), and Pap 3′-UTR del (sense, TCGAGTCTACTCTTTTACCAAGTATACTATAAATTAGAGCTT; antisense, CTAGAAGCTCTAATTTATAGTATACTTGGTAAAAGAGTAGAC).

Chinese hamster ovary (CHO) cells were seeded at a density of 4 × 10^4^ cells per well in 96-well plates. When the cells reached 60 to 70% confluence, the reporter vector pmirGLO-PAP, pmirGLO-PAP-Mut, and pmirGLO-PAP-Del were co-transfected with the miR-125b-5p mimic or negative control (NC) into the CHO cells. GenEscort II (Huiji, Nanjing, China) was used to mediate the transfection procedure according to the manufacturer’s protocol. The transfection efficiency was normalized by the activity of Renilla-luciferase. The luciferase assay was performed with the Dual-Luciferase reporter assay system (Promega, Madison, WI, USA).

### 4.9. RNA Extraction and qRT-PCR

The total RNA was extracted from the testicular tissues using a Trizol reagent. The RNA concentration was determined using the NanoDrop 2000. The total RNA (2 μg) was reverse-transcribed to cDNA using the Moloney Murine Leukemia Virus reverse transcriptase (Promega, Madison, WI, USA) with OligodT18 (IR), or the Universal Adaptor Primer (miRNAs) from the One Step PrimeScript^®^ miRNA cDNA Synthesis Kit. After 1 h of incubation at 42 °C and 10 min of deactivation at 75 °C, the reaction mixes were used as the templates for PCR. Real-time PCR was performed with standard protocols on the STRATA-GENE Mx3005P sequence detection system. The PCR mixture contained 2 μL of cDNA, 10 μL of 2 × SYBR Green PCR Master Mix, 0.5 μL of each primer, and some water to make up the final volume to 20 μL. The reaction was performed in a 96-well optical plate at 95 °C for 1 min, followed by 35 cycles of 95 °C for 15 s, optimal reannealing temperature for 15 s and 72 °C for 40 s. All of the reactions were run in duplicate and a negative control (NC) without template was included for each gene. The primers of RT-PCR are shown in Appendix A. The primers were designed based on the sequence of each gene by using Premier 5.0 (PREMIER Biosoft, Palo Alto, CA, USA).

### 4.10. Sperm mtDNA Integrity

The mitochondria of the sperms were extracted using the QIAamp^®^ DNA Mini Kit. The mtDNA integrity was carried out following a previous study [46]. Briefly, long PCR amplification was applied to 8220 and 8140 bp out of the 16 kb mitochondrial genome. Primers of the mtDNA fragment are shown in Table 2, and were designed and synthesized by Sangon (Sangon Co., LTD, Shanghai, China). The amplification conditions were as follows: 30 cycles of denaturing (98 °C for 30 s), annealing (55 °C for 15 s), and extension (68 °C for 9 min). The PCR products were examined by electrophoresis with 1% agarose gel, and visualized using 4S Red Plus Nucleic Acid Stain (Sangon Co., LTD, Shanghai, China).

### 4.11. Sperm mtDNA Mutations of mt-Cytb and mt-ATP6

The mutations of mt-Cytb and mt-ATP6 were evaluated by PCR and sequence analysis by Mao et al. [47]. Table 3 shows the primer sequences, which were synthesized by Sangon (Sangon Co., LTD, Shanghai, China). The PCR amplification conditions were as follows: 94 °C for 5 min, followed by 30 cycles at 94 °C for 30 s, 54 °C for 15 s (mt-Cytb, mt-ATP6), and 72 °C for 80 s (mt-Cytb, mt-ATP6). The mutations in the PCR-amplified products were analyzed using blast analysis.

### 4.12. siRNA Transfection

Three siRNAs oligos targeting the PAP and negative control were designed and synthesized from Guangzhou RiboBio Co (Appendix A). The SiRNA oligos were transfected into the TM3 cell using Lipofectamine 2000 (Invitrogen), following the manufacturer’s instructions, and the most effective siRNA was chosen for subsequent analysis. After transfection with the most effective siRNA or NC, the levels of testosterone in the supernatant were measured.

### 4.13. Cell Isolation and Purification

The Leydig cells (LC) were isolated according to the protocol described previously [48,49]. Briefly, the testes from three week-old WT mice were collected and washed using PBS. Then, the tunica-free testes were incubated with 0.25 mg/mL collagenase type I (Sigma, St. Louis, MO, USA) at 32 °C for 10 min with a gentle swing. The suspensions were allowed to settle naturally and were filtered through 75 mm copper meshes to separate the interstitial cells and seminiferous tubules. The interstitial cells were cultured in F12/DMEM (Life Technologies, Inc., Grand Island, NY, USA) with 100 U/mL penicillin, 100 mg/mL streptomycin, and 10% FBS (Gibco, Grand Island, NY, USA). After 15 min, the non-adherent and loosely adherent cells were collected and resuspended by F12/DMEM with 10% FBS, 100 U/mL penicillin, 100 mg/mL streptomycin, and cultured at 5% CO2 at 34 °C.

The seminiferous tubules were re-suspended in PBS, collagenase type I at 32 °C for 15 min to remove the testicular peritubular myoid cells (TPC). The tubules were broken into small pieces and incubated with 1 mg/mL hyaluronidase (Sigma) at 32 °C for 10 min with gentle pipetting to separate germ cells (GCs) and Sertoli cells (SCs) [50]. The suspensions were cultured with F12/DMEM at 32 °C for 6 h. The GCs were recovered by collecting the non-adherent cells. The SCs were cultured for an additional 48 h and then treated with a hypotonic solution (20 mM Tris; pH 7.4) for 2 min to remove GCs [51].

### 4.14. Statistical Analysis

All of the experiments included at least five replicates per group. The data were evaluated by Student’s *t*-test or one-way ANOVA analysis, and the differences between the groups were considered statistically significant at *p* < 0.05. All of the statistical analyses were performed with PASW Statistics 18 software.

## Figures and Tables

**Figure 1 ijms-20-00148-f001:**
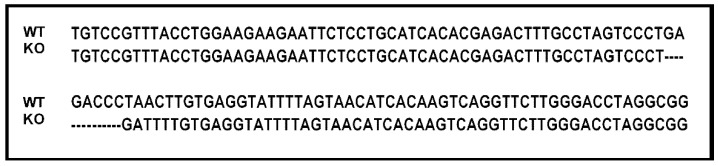
The sequences of wild type (WT) mice and miR-125b-2 knockout (KO) homozygous mice were compared using bio XM Software.

**Figure 2 ijms-20-00148-f002:**
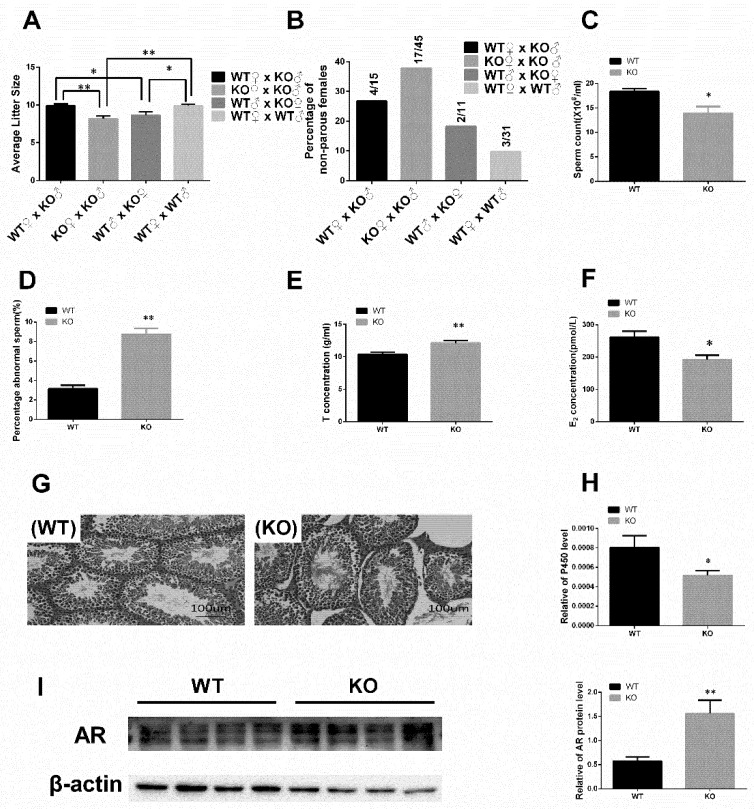
miR-125b-2 knockout causes different reproduction phenotypes. (**A**,**B**) The effect of miR-125b-2 knockout on the fertility of male mice. WT and KO males were examined in the following four combinations: WT♀ × KO♂, KO♀ × KO♂, WT♂ × KO♀, and WT♀ × WT♂. The pups (**A**) and litter rate (**B**) were counted (*n* = 15–20). (**C**,**D**) Effect of miR125b-2 on sperm count (**C**) and percentage of abnormal sperms (**D**) (*n* = 8). T (testosterone) concentration (**E**) and E_2_ (estradiol) concentration (**F**) in nine-week-old WT and KO mice (*n* = 8). (**G**) Microscopic observations of seminiferous tubules in WT and KO mouse testes: a—wild-type FVB/NJ (FVB) male with normal spermatogenesis; b—KO FVB/NJ male (recipient strain). (**H**) P450 expression in mouse testis in KO mice (*n* = 8). (**I**) Characterization of androgen receptor (AR) protein and gray-scale scanning analyses in WT and KO mouse testis (*n* = 4). All of the data were expressed as mean ± SEM. * *p* < 0.05, ** *p* < 0.01.

**Figure 3 ijms-20-00148-f003:**
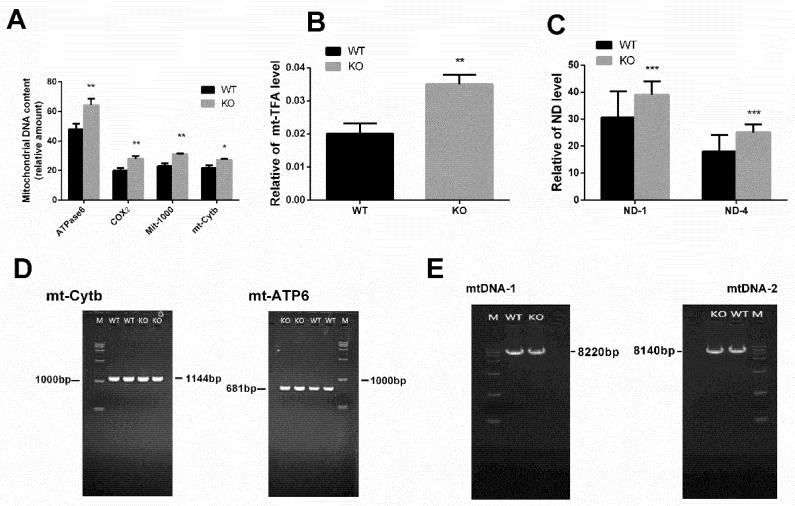
miR-125b-2 affected mtDNA’s copy number and integrity. (**A**) MtDNA copy number in the sperm of the KO and WT mice. (**B**) The relative expression of mitochondrial transcription factor A (mt-TFA) (*n* = 8). (**C**) Quantifications of ND1 and ND4 mRNA levels in sperm. (**D**) PCR amplification products of mtCytb (left) and mtATP6 (right) in the sperm of KO and WT mice. M: DL2000 DNA marker. (**E**) 8220 bp (left) and 8140 bp (right) of the mtDNA amplification products in the sperm of KO and WT mice using long PCR. M: DL15000 DNA marker. The asterisk indicates a statistically significant difference at * *p* < 0.05, ** *p* < 0.01, *** *p* < 0.001.

**Figure 4 ijms-20-00148-f004:**
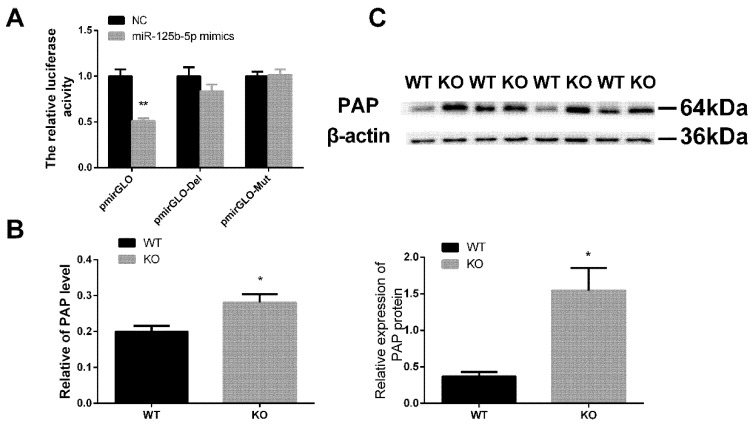
Papolb (PAP) is the target of miR-125b-5p. (**A**) PmirGLO dual-luciferase reporter vectors analysis (*n* = 8). Relative luciferase activities were calculated by firefly luminescence/Renilla luminescence. (**B**) The PAP expression in mouse testis after knockout miR-125b-2 (*n* = 8). (**C**) Characterization of the PAP protein and gray-scale scanning analyses in WT and KO mouse testis (*n* = 4). Values are the mean ± SEM. * *p* < 0.05, ** *p* < 0.01.

**Figure 5 ijms-20-00148-f005:**
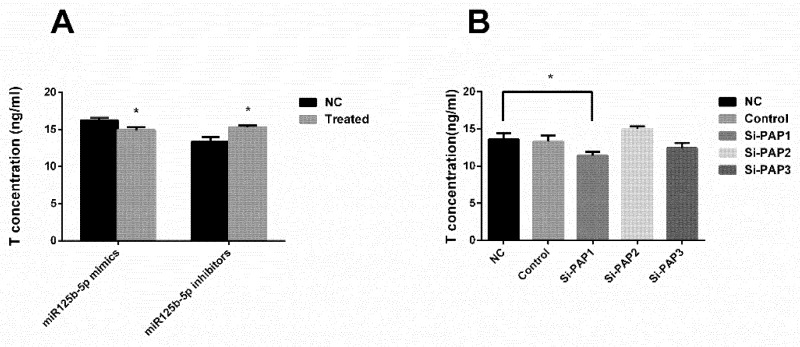
MiR-125b-5p and PAP affected TM3 cell secreted testosterone. (**A**) TM3 cells were transfected with miR-125b-5p mimics/NC/inhibitor/iNC. (**B**) TM3 cells were transfected with different kinds of PAP siRNAs. The T (testosterone) concentration of the cell supernatants were determined (*n* = 6). All of the data are expressed as mean ± SEM. * *p* < 0.05.

**Figure 6 ijms-20-00148-f006:**
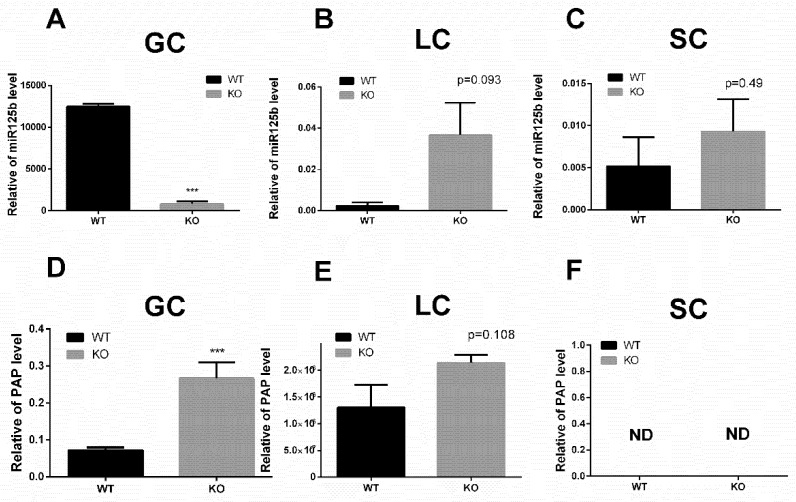
The Relative Expression of miR-125b-2 and PAP mRNA in Leydig cells (LC), Sertoli cells (SC), and germ cells (GC). (**A**) The relative level of miR-125b in GC. (**B**) The relative level of miR-125b in LC. (**C**) The relative level of miR-125b in SC. (**D**) The relative of mRNA level PAP in GC. (**E**) The relative mRNA level of PAP in LC. (**F**) The relative mRNA level of PAP in SC (*n* = 5). ND—not detected. Values are the mean ± SEM. *** *p* < 0.001.

**Table 1 ijms-20-00148-t001:** Body and testicular tissues of wild type (WT) and knockout (KO) mice were weighted (*n* = 8). BW—body weight; TW—testis weight; *n* = 8. All of the data were expressed as mean ± SEM. * *p* < 0.05.

Group	BW (g)	TW (mg)
WT	29.12 ± 0.78	166.50 ± 8.43
KO	28.09 ± 0.89	167.50 ± 6.38

**Table 2 ijms-20-00148-t002:** Primer sequences and their corresponding PCR product sizes for long PCR.

Gene	Primer Sequences	Accession No.	Product Sizes (bp)
mtDNA-1	F:GTTAATGTAGCTTAATAACAAAGCAAAGC	NC_005089.1	8220
	R:TAGTTGGGTAGTAGGTGTAAATGTATGTG		
mtDNA-2	F:ATTGGATCAACAAATCTCCTAGG	NC_005089.1	8140
	R:TTGTTAATGTTTATTGCGTAATAGAGTATG		

**Table 3 ijms-20-00148-t003:** Primer sequences and their corresponding PCR product sizes for mutations of mt-Cytb and mt-ATP6.

Gene	Primer Sequences	Gene ID	Product Sizes (bp)
mt-Cytb	F:ATGACAAACATACGAAAAACACA	R17711	1144
R:ATGGATATAATTTTAGTATTTTGTCTTCGA		
mt-ATP6	F: ATGAACGAAAATCTATTTGCCTC	17705	681
R:TTATGTATTATCATGTAGATATAGGCTTACTAGGA

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
