# Peer review of "MiR-125b-2 Knockout in Testis Is Associated with Targeting to the PAP Gene, Mitochondrial Copy Number, and Impaired Sperm Quality"

_ijms, 2019, doi:10.3390/ijms20010148_

Reviewer 1 Report

Although the paper was basically well written, there are many tiny parts that should be corrected..

All significant P-values (P<0.05,P<0.01) should be denoted explicitly with their actual values, e.g., P=0.042.

Use × instead of X for "times".

Characters and numeric in Fig. 2 and S2 are too fine to read.

L92 "T and E2" -> What are "T and E2"?

L93 ((Figure 2E-F). -> (Figure 2E-F).

L124 p<0.05 -="">What kind of software was used? DESeq2? Were these P-values corrected with considering multiple comparisons?

L126 "the most significant differences" -> What is exact criterion? You used P-values as well as fold change. It is not obvious what "most significant" means.

Author Response

Thank you for your valuable suggestions given to us. These suggestions are valuable and helpful for improving our paper and provide beneficial guidance to our studies. We have studied the comments carefully and have made corrections which we hope to meet your approval. The main corrections we made in the paper and are as following:

Comment 1: All significant P-values (p<0.05, p<0.01) should be denoted explicitly with their actual values, e.g., P=0.042.

Response 1: Yes, this is a good suggestion. All significant P-values (p<0.05, p<0.01) has been denoted explicitly with their actual values, please see the revised version. Many thanks. 

Comment 2: Use × instead of X for "times".

Response 2: Yes, this is a good suggestion. “X” has been corrected by using “×”, please see lines 80-85, lines 107-108 and Figure 2 in the revised version. Many thanks.

Comment 3: Characters and numeric in Fig. 2 and S2 are too fine to read.

Response 3: Yes, this is a good suggestion. Figure 2 and S2 have been re-edited. We have improved the size resolution of Fig 2, please see lines 105 in the revised version. As for Figure S2, we have divided this figure into three pictures, Figure S2, Figure S3 and Figure S4, deleted the original Figure S2B because its very low-resolution, please see lines 580-588 in the revised version. Many thanks.

Comment 4: L92 "T and E2" -> What are "T and E2"?

Response 4: Yes, we re-defined T and E2, T is an abbreviation of testosterone, please see line 92 in the revised version. E2 is an abbreviation of estradiol please see line 93 in the revised version. Many thanks.

Comment 5: L93 ((Figure 2E-F). -> (Figure 2E-F).

Response 5: Yes, this is a clerical error and we have corrected it, please see lines 94-95 in the revised version. Many thanks.

Comment 6: L124 p<0.05 -="">What kind of software was used? DESeq2? Were these P-values corrected with considering multiple comparisons?

Response 6: Yes, this is a good suggestion. The mRNA of significant differences genes (p<0.05) were screened by Ballgown [17] and the P-value was multiple testing correction by false discovery rate (FDR)[18]. We have changed the description in the lines 128-130 of the revised version. Many thanks.

References:

17.Frazee, A. C.; Pertea, G.; Jaffe, A. E.; Langmead, B.; Salzberg, S. L.; Leek, J. T., Flexible isoform-level differential expression analysis with Ballgown. Biorxiv 2014.

18.Benjamini, Y.; Hochberg, Y., Controlling The False Discovery Rate - A Practical And Powerful Approach To Multiple Testing. Journal of the Royal Statistical Society 1995, 57, (1), 289-300.

Comment 7: L126 "the most significant differences" -> What is exact criterion? You used P-values as well as fold change. It is not obvious what "most significant" means.  

Response 7: Yes, this is a good suggestion. The criterion of the most significant differences in our study is based on the P-value. Firstly, we selected DEGs with fold change≥2 and p<0.05. Then, according to the order of p-value from small to big, the top 10 genes with up-regulation and down-regulation were selected respectively (Supplemental File Table S2 and S3). To determine the function of DEGs, we mapped them both in the GO database (Supplemental File Figure S2) and KEGG Pathway database (Supplemental File Figure S3). We restated these criterions; please see lines 133-137 in the revised version. Many thanks.

Reviewer 2 Report

The authors tried to find a connection between microRNA and KO sperm quality. It seems a very good study but loses itss importance very much due to poor English. The whole ms needs linguistic improvement. For example: The first sentence in the Abstract: MicroRNAs can cause male infertility....? the definition of microRNAs is different. Secondly, although the authors found that in the KO animals the sperm count was reduced through tthe microRNA pathway at the end they conclude that microRNA used in the study can be used for treating male infertility.

The authors should correct these paragraphs

Author Response
The authors tried to find a connection between microRNA and KO sperm quality. It seems a very good study but loses its importance very much due to poor English. The whole manuscript needs linguistic improvement. For example: The first sentence in the Abstract: MicroRNAs can cause male infertility....? The definition of microRNAs is different. Secondly, although the authors found that in the KO animals the sperm count was reduced through the microRNA pathway at the end they conclude that microRNA used in the study can be used for treating male infertility.
The authors should correct these paragraphs.
Response: Yes, thank reviewer #2 for your encouragement and valuable suggestions given to us. These suggestions are valuable and helpful for improving our paper and providing beneficial guidance to our studies. For example, in the abstract, as the reviewer said, the definition of miRNA may cause ambiguity to the purpose of the whole article, we deleted the first sentence “MicroRNAs can cause male infertility…”. Please see line 10 in the revised version. In the conclusion, we restated conclusion that “…could be a potential drugs and diagnostic target for male infertility.”, changed “the therapeutic target” to “drugs and diagnostic target”. Please see line 32 and 63 in the revised version. In the whole article, the knockout of miR-125b-2 could reduce sperm count and increase both the percentage of abnormal sperms (lines 87-90 in the revised version) and serum testosterone content (lines 93-95 in the revised version). Moreover, the transcriptome analysis of the mice testicular showed that PAP, a testis specific expressing gene, increased significantly in the KO mice testis, then qRT-PCR, western blotting and Luciferase assay collectively validated that miR-125b-2 decreased testosterone by directly targeting PAP (lines 182-187 in the revised version). These results indicated that miR-125b-2 had a positive influence on the reproductive performance of animal by directly targeting on PAP gene and could be a potential drugs and diagnostic target for male infertility. We re-edited the whole manuscript and improved the linguistic quality of the paper. Please see the revised version. Many thanks.

Round  2

Reviewer 2 Report

The authors made a revision of the ms and to some extent the authors answered to the reviewer's suggestions. Moreover, the authors may elaborate about the potential pathway of miRNA action and to extrapolate some of the findings to humans.

Author Response

Comment:The authors made a revision of the ms and to some extent the authors answered to the reviewer's suggestions. Moreover, the authors may elaborate about the potential pathway of miRNA action and to extrapolate some of the findings to humans.
Response: Yes, this is a good suggestion. Our study confirmed the potential mechanism of miR-125-2 action that miR-125b-2 regulated testosterone secretion by directly targeting PAP and increased sperm mtDNA copy number to affect semen quality (please see the description in lines 28-30 and 300-302). In this study, we found that miR-125b and PAP were mainly expressed in the germ cells, which was consistent with others’ reports [30, 33]. MiR-125b-2 knockout caused infertility and increased significantly the sperm mitochondria DNA (mtDNA) copy number. Meanwhile, miR-125b targeting PAP gene was first identified, when miR-125b-5p over-expression, the secretion of testosterone can be down-regulated in TM3 cell by targeting PAP (p=0.021) (please see the description in lines 291-299). PAP, as a testis specific expressing gene, affects the process of spermatogenesis, but the study of PAP gene is not too much (please see the description in lines 263-279), especially, the function of miR-125b targeting PAP in vivo. For example, what pathways are taken by this target relationship to regulate testosterone secretion and sperm mitochondrial function remain still unclear, such as NF-kappa B, STAT3, mTORC1, p38MAPK/ERK and TGF-β signaling pathway, when PAP gene conditioned knockout or specific overexpression in mice. So, it also needs further research and exploration. Even better, our colleagues are starting to study these areas. Only in this way can we get a more detailed pathway to miR-125b function that was associated with targeting to PAP gene, mitochondrial copy number and impaired sperm quality. Of course, these studies in mice are also very good reference for human infertility (please see the description in lines 30-32, 62-63, 300-305). In addition, we have also made some minor changes. Many thanks.

References:
30.   Kashiwabara, S.; Noguchi, J.; Zhuang, T.; Ohmura, K.; Honda, A.; Sugiura, S.; Miyamoto, K.; Takahashi, S.; Inoue, K.; Ogura, A., Regulation of spermatogenesis by testis-specific, cytoplasmic poly(A) polymerase TPAP. Science 2002, 298, (5600), 1999.
33. Kashiwabara, S.; Yamagata, K. N. J.; Fukamizu, A.; Baba, T.; Zhuang, T. G., Identification of a novel isoform of poly(A) polymerase, TPAP, specifically present in the cytoplasm of spermatogenic cells. Developmental Biology 2000, 228, (1), 106-115.
